# An Introduced RNA-Only Approach for Plasmid Curing via the CRISPR-Cpf1 System in *Saccharomyces cerevisiae*

**DOI:** 10.3390/biom13101561

**Published:** 2023-10-23

**Authors:** Bo-Chou Chen, Yu-Zhen Chen, Huan-Yu Lin

**Affiliations:** 1Bioresource Collection and Research Center, Food Industry Research and Development Institute, Hsinchu 300, Taiwan; bcc@firdi.org.tw; 2Department of Food Science and Technology, Hungkuang University, Taichung 433, Taiwan; u108p227@gmail.com

**Keywords:** CRISPR-Cpf1, plasmid-curing, genome-editing, *Saccharomyces cerevisiae*

## Abstract

The CRISPR-Cas system has been widely used for genome editing due to its convenience, simplicity and flexibility. Using a plasmid-carrying Cas protein and crRNA or sgRNA expression cassettes is an efficient strategy in the CRISPR-Cas genome editing system. However, the plasmid remains in the cells after genome editing. Development of general plasmid-curing strategies is necessary. Based on our previous CRISPR-Cpf1 genome-editing system in *Saccharomyces cerevisiae*, the crRNA, designed for the replication origin of the CRISPR-Cpf1 plasmid, and the ssDNA, as a template for homologous recombination, were introduced for plasmid curing. The efficiency of the plasmid curing was 96 ± 4%. In addition, we further simplified the plasmid curing system by transforming only one crRNA into *S. cerevisiae*, and the curing efficiency was about 70%. In summary, we have developed a CRISPR-mediated plasmid-curing system. The RNA-only plasmid curing system is fast and easy. This plasmid curing strategy can be applied in broad hosts by designing crRNA specific for the replication origin of the plasmid. The plasmid curing system via CRISPR-Cas editing technology can be applied to produce traceless products without foreign genes and to perform iterative processes in multiple rounds of genome editing.

## 1. Introduction

Circular plasmids have been studied in prokaryotes for over 60 years [1], and were first discovered in yeast in 1971 [2]. Plasmids may be independently replicated genetic molecules in addition to the chromosome, but the presence of plasmids is not essential for the viability of microorganisms. Plasmids are important molecular tools that are used to delete, insert or modify target genes in the development of genetic biotechnology. Plasmids can autonomously replicate through self-control, and the origin of replication (*ori*) is the characteristic of each replicon [3,4,5]. Yeast integrating plasmids (YIp), yeast centromere plasmids (YCp), and yeast episomal plasmids (YEp) are commonly used plasmids in *Saccharomyces cerevisiae*. YIp lacks replication initiation sites but is stabilized when integrated into the chromosome [6]. YCp contains a yeast centromere (CEN) and an autonomously replicating sequence (ARS), and it has high mitotic stability but low copy numbers. YEp contains a 2 µ plasmid replication origin and a partitioning locus, and it has high copy number characteristics but is unstable [7,8].

Clustered regularly interspaced short palindromic repeats-CRISPR associated protein (CRISPR-Cas) is an adaptive immune system present in some bacteria or archaea, and it has been developed as a genome-editing tool in a wide range of organisms including bacteria and eukaryotes [9]. The CRISPR-Cas system has been widely used for genome editing due to its convenience, simplicity and flexibility [10]. The CRISPR-Cas9 system from *Streptococcus pyogenes* has been widely used for metabolic engineering in *S. cerevisiae*. Recently, a novel and efficient genome editing tool, CRISPR-Cpf1, has been developed in *S. cerevisiae* [11]. Cpf1 (Cas12a) and Cas9 are both RNA-guided endonucleases that are members of the class 2 CRISPR-Cas systems [12,13]. In contrast to Cas9, Cpf1 utilizes a T-rich protospacer adjacent motif (PAM) rather than a GC-rich PAM [14,15], is guided by a single CRISPR RNA (crRNA) rather than by a chimeric single guide RNA (sgRNA) and a trans-activating CRISPR RNA (tracrRNA) [16,17], and cleaves DNA into sticky ends rather than into blunt ends [13]. Recent studies have used CRISPR-Cas9 or CRISPR-Cpf1 systems for genome editing in *S. cerevisiae* using plasmids carrying CRISPR elements.

Plasmid curing is the process by which plasmids are removed from cell populations. Eliminating plasmids and achieving plasmid-free cells is a major challenge [18]. Plasmid curing can be achieved by inhibition of various stages of the plasmid replication process, including replication, partition, or transfer [19]. Any plasmid-free cells have arisen due to occasional failures of replication, multimer resolution, or partitioning [20]. Traditional methods of plasmid-curing are based on prolonged growth under stressful conditions, such as elevated temperature or the addition of DNA intercalating agents to interfere with plasmid replication. Because the cells that do not contain plasmids are rapidly enriched under non-selective conditions, plasmids can be cured naturally through cell division by continuous subculture and by screening for the loss of plasmid. The temperature-sensitive replication origin could be generated for an efficient exclusion of vectors [21,22]. However, these are usually laborious and time-consuming processes. The application is limited to large size, temperature restrictions, low copy numbers, and little variety [18]. Therefore, it is necessary to develop an efficient method for specific plasmid elimination.

Recently, the CRISPR-Cas system has been explored as a method for plasmid curing. The advantages of using CRISPR-Cas systems for plasmid curing are clear. The system can be designed to target specific plasmid genes, the double-stranded breaks introduced in the process can reduce the stability of the plasmid, and result in plasmid loss [23]. The inducible plasmid curing system is constructed based on CRISPR-Cas9 genome editing technology in *Escherichia coli*. The system is composed of five elements, including Cas9 constitutively expressing cassette, guide RNA (gRNA) expression plasmid, λ Red recombineering system, donor template DNA, and inducible plasmid curing system. When induced by arabinose, the gRNA targeting the *bla* gene is expressed, leading Cas9 to cleave the gRNA plasmid and resulting in plasmid elimination [24]. The multigene editing CRISPR-Cas9 system is demonstrated in *E. coli* using the two-plasmid system, pCas and pTarget. The plasmid pCas9 contains the *cas9* gene with a native promoter and an arabinose-inducible sgRNA guiding Cas9 to the pMB1 replication origin of pTarget, and pCas9 also contains the temperature-sensitive replication element repA101 for self-curing [25].

In the previous study, we established a CRISPR-Cpf1 genome-editing system in *S. cerevisiae*, using a visual recognition model that presents red colonies when the *ADE2* gene (encoding phosphoribosylaminoimidazole carboxylase) is deleted [26]. Furthermore, we also constructed the edited strains with a double-gene deletion, including the genes encoding the trehalose degradation enzyme (Nth1) and the membrane chaperone (Hsp12) by the CRISPR-Cpf1 system in *S. cerevisiae*. Deletion of *NTH1* and *HSP12* increases the freeze–thaw resistance of baker’s yeast in bread dough [27]. However, the edited strains containing the CRISPR plasmids are difficult to apply in the food industry. Whether the strain is intended for the next round of genome editing or for use in the food industry, the elimination of plasmids from the bacteria is an essential step. In this study, based on the CRISPR-Cpf1 genome-editing system, crRNA targeting to the replication origin CEN/ARS of the CRISPR plasmid was transformed into edited strains, resulting in Cpf1 endonuclease cleavage of the plasmid and plasmid cure. The CRISPR-mediated plasmid-curing system was generated by introducing only a single crRNA fragment, and it can be used for multi-round CRISPR-Cpf1 genome editing in *S. cerevisiae*.

## 2. Materials and Methods

### 2.1. Strains, Media and Growth Conditions

The *S. cerevisiae* wild-type strain BCRC 21447 used in this study was obtained from the Bioresource Collection and Research Center. The Δ*ade2*, Δ*nth1*, Δ*hsp12*, and Δ*nth1*/Δ*hsp12* strains were constructed from the previous study [26,27]. *S. cerevisiae* was grown at 30 °C in a YPD medium (1% yeast extract, 2% peptone, and 2% glucose)(BD, Sparks, MD, USA), and 200 mg/L G418 (Sigma-Aldrich, St. Louis, MO, USA) was added for the selection.

### 2.2. Plasmid Construction and crRNA Design

The plasmid pBCscADE2 used in this study was constructed by the previous study [26], which was applied to express the *Fn*Cpf1 cassette, the crRNA sequence crADE2 targeting the gene for *ADE2* gene deletion and the KanMX cassette expressing a resistance gene for G418 (geneticin) (Figure 1a). The crRNA sequences (crCEN1 and crCEN2) targeting the replication origin CEN/ARS of the plasmid were designed from the CHOPCHOP website (http://chopchop.cbu.uib.no/ accessed on 27 September 2023), and the synthesized single-stranded DNA (ssDNA), as a homologous DNA template (CEN HA), contained 50 bp of DNA upstream and downstream of the replication origin CEN/ARS (Table 1). The crRNAs and oligonucleotides were purchased from Integrated DNA Technologies Int (IDT, Coralville, IA, USA).

### 2.3. Transformation and Plasmid-Curing in Yeast

All yeast transformations were performed using the modified lithium acetate protocol by electroporation as previously described [28]. *S. cerevisiae* was incubated at 30 °C overnight until it reached the stationary growth phase. Subsequently, 50 μL of a bacterial solution was introduced into a 20 mL YPD medium and incubated at 30 °C overnight until the OD_600_ reached a range of 0.5 to 1.2. The bacterial solution was then centrifuged, the culture medium was removed, and the bacterial cells were washed once with ice-cold water. In succession, a solution consisting of 0.1 M lithium acetate (LiAc) containing 1 M sorbitol (Sigma-Aldrich, St. Louis, MO, USA), 10× TE buffer, and 1 M LiAc (Sigma-Aldrich, St. Louis, MO, USA), along with 1 M dithiothreitol (DTT) (Sigma-Aldrich, St. Louis, MO, USA), was added at 30 °C. The cells were subjected to two additional washes with ice-cold 1 M sorbitol. Finally, the bacterial cells were suspended in 1 M sorbitol, completing the preparation of competent cells.

In the construction of the *ADE2*-deleted *S. cerevisiae* strains, 500 ng pBCscADE2 and 1000 ng donor DNA were introduced into the competent cells, and subjected to electroporation at 1.25 kV. Subsequently, 1 mL YPD medium and 1 mL 1 M sorbitol were added, and the cells were incubated at 30 °C with shaking for 2–48 h to repair the cells. The resulting transformants were sub-cultured on YPD medium supplemented with antibiotics G418 at 30 °C for 48–72 h [26]. For the construction of the plasmid-cured *S. cerevisiae* strains, 100 nmol crRNA fragments (crCEN1 or crCEN2) and 1000 ng ssDNA template (CEN HA) were introduced into the competent cells. Following recovery, the transformants were sub-cultured on the YPD medium, and a single colony was sub-cultured on both YPD mediums with and without 200 mg/L G418 supplementation. The efficiency of plasmid-curing was calculated as the percentage of non-growing colonies out of the total number of colonies in YPD plates containing G418 antibiotics. The strains were confirmed for plasmid curing by colony PCR analysis using F1-F/KX-R primer pairs, which are specific for the plasmid backbone DNA (Table 1).

### 2.4. Statistical Analysis

All the experiments were performed individually at least three times, and the data reported were mean values ± SD. The differences were confirmed by Student’s *t*-test. Differences were considered statistically significant at *p* < 0.05.

## 3. Results

### 3.1. Design a Plasmid-Curing System Based on the Cleavage of the Replication Origin

The *ADE2*-deleted *S. cerevisiae* strains were constructed using the CRISPR-Cpf1 genome editing system by transforming an ssDNA donor template and a plasmid pBCscADE2, containing crRNA and Cpf1 expression cassettes. The edited strains appeared as red colonies, because deletion of the *ADE2* gene resulted in adenine auxotrophy [26]. To obtain an edited strain ready for the next round of editing, the plasmid used for genome editing should be removed. Two sets of crRNA sequences, crCEN1 and crCEN2, were designed to target the replication origin CEN/ARS of the plasmid. An ssDNA donor, CEN HA, homologous to the replication origin CEN/ARS over a region extending 50 bp upstream and 50 bp downstream, was designed (Figure 1a).

A total of 100 nmol of crCEN1 or crCEN2 was transformed with 1 μg CEN HA into the Δade2 strain by electroporation, and the transformants were further incubated for 48 h at 30 °C on YPD medium for recovery. After recovery, the transformants were incubated on YPD medium, and then a single colony was sub-cultured on both YPD mediums with and without G418 antibiotics. The results showed that most of the colonies were sensitive to G418 antibiotics, indicating that the plasmid-curing system works well (Figure 1b). Colony PCR was also applied to verify the presence of the plasmid, and the presence of pBCscADE2 was undetectable in cured strains with a detection limit of 5 fg of target DNA (Figure 1c). Compared to two design crRNAs for the plasmid replication origin, the curing efficiency was 96 ± 4% using crCEN1, and significantly higher than 57 ± 6% using crCEN2 (*p* < 0.01) (Figure 1d). When there is no replication origin in the plasmid, it becomes unable to replicate. Therefore, disruption of the replication origin of CRISPR plasmids leads to the plasmid elimination in *S. cerevisiae*.

### 3.2. CRISPR Plasmids Can Be Cured with Only a Single crRNA

To further simplify the plasmid curing system, only crCEN1 was transformed into the Δ*ade2* strain without the donor DNA CEN HA, and the curing efficiency remained at 73 ± 4% (Figure 2a). This indicated that once the replication origin sequence of the plasmid was broken, the plasmid could be eliminated [29], and the G418-sensitive clones have successfully removed the CRISPR plasmid. By introducing only crCEN1, the CRISPR plasmid could be eliminated successfully. Without introducing the ssDNA, cleavage of the replication origin sequence of the CRISPR plasmids by the complex of crRNA/Cpf1 disrupts the progression of replication. Based on the above results, we have obtained an efficient and low-cost crRNA-only plasmid-curing system, and the donor DNA was not essential for plasmid curing.

### 3.3. The Economic and Efficient Plasmid-Curing System

To optimize the plasmid-curing system, the different concentrations of crCEN1, including 1 nmol, 10 nmol, and 100 nmol, were investigated. The results showed that the curing efficiency of 10 nmol and 100 nmol crCEN1 was 73 ± 6% and 73 ± 4%, respectively. At 1 nmol crCEN1, the curing efficiency was significantly lower at 57 ± 5% compared to 10 nmol and 100 nmol (Figure 2b). It suggested that the 10 nmol of crCEN1 was effective in the plasmid-curing system. Furthermore, different recovery periods for electroporation, including 2 h, 5 h, and 48 h, were investigated with 10 nmol crCEN1 transformation conditions. The results indicated that the curing efficiency was highest for 5 h incubation after electroporation at 71.0 ± 4.3% (Figure 2c). A convenient and efficient plasmid-curing system was constructed with 10 nmol crCEN1 transformation and a 5 h recovery period, and the curing efficiency was approximately 70%.

### 3.4. The Genome Edited and the Plasmid-Free Yeast Strains

In the previous study, we generated the freeze–thaw resistant *S. cerevisiae* strains with *NTH1* deleted, *HSP12* deleted, and *NTH1*/*HSP12* double deleted using the CRISPR-Cpf1 system [27]. However, the edited strains contain CRISPR plasmids, and strains containing foreign genes are difficult to apply in the food industry. To investigate plasmid-curing system, the Δ*nth1*, Δ*hsp12*, and Δ*nth1*/Δ*hsp12* strains were transformed with 10 nmol crCEN1. The plasmid cured strains were selected by YPD medium without antibiotics and confirmed by PCR analysis (Figure 3).

We established a multiple-round platform for genome editing in *S. cerevisiae*. First, the plasmid, which contains crRNA specific for the targeted sequence and Cpf1 expression cassettes and the homologous templates (ssDNA), are introduced into *S. cerevisiae* strains for genome editing. A crRNA/Cpf1 complex is formed in yeasts, and crRNA guides the crRNA/Cpf1 complex to target DNA. The target DNA sequence will be cleaved by the crRNA/Cpf1 complex, and the DNA repair system will be induced to repair DNA breaks (Figure 4a). The editing strains for specific target gene editing are screened and selected (Figure 4b). The crRNA fragments specific for the replication origin of the CRISPR plasmid are introduced into *S. cerevisiae* strains containing the CRISPR plasmids. The introduced crRNA could form a complex with Cpf1 inside the strains. The replication origin sequence is cleavage by the crRNA/Cpf1 complex to eliminate the CRISPR plasmid (Figure 4c). Finally, the plasmid-free editing strain is harvested (Figure 4d). The next round of genome editing can be carried out. This study demonstrates the plasmid-curing system for developing safe host strains and constructing food-grade *S. cerevisiae* without residual antibiotic markers. Multi-round CRISPR-Cpf1 genome editing in *S. cerevisiae* can be performed by using the simple and effective plasmid-curing system.

## 4. Discussion

In this study, we established a plasmid-curing system using CRISPR-Cpf1 technology in *S. cerevisiae*. Based on the edited strains containing CRISPR-Cpf1 plasmid, the edited strains, which were introduced with only one crRNA fragment, could remove the CRISPR plasmid, and the efficiency of plasmid curing was about 70%. The next round of CRISPR genome editing steps will be available. The plasmid-curing system can be used for multi-round CRISPR-Cpf1 genome editing in *S. cerevisiae*.

Under non-selective conditions, plasmids can be cured naturally through cell division by continuous subculture and by screening for the loss of plasmid. Initially, we serial sub-cultured the edited strains on YPD medium without any antibiotics. However, no plasmid-cured cells were obtained. Then, we tried to develop the CRISPR system to cure the plasmid. In this study, only a free crRNA fragment was introduced, and the crRNA and Cpf1 complex could target the replication origin of the plasmid and eliminate the plasmid.

In a previous study, a Cas-3P system was developed for sequential genome engineering, and involved plasmid curing strategies. This system allowed immediate, multiplexed, and sequential gene targeting in *S. cerevisiae*. The three marked gRNA plasmid backbones could be recurred in a P1–P2–P3 order for sequential gene targeting, without elimination of the gRNA plasmid by induction in each round. Three sites of CS1, CS2, and CS3 were constructed in P1, P2, and P3, and it could be used for plasmid curing by constitutively expressed gRNA_CS1 in P2, gRNA_CS2 in P3, and gRNA_CS3 in P1, respectively. The gRNA plasmid generated in the previous round could be eliminated by the Cas9 coupled with the gRNA expressed by the plasmid generated in the next round [29]. The edited strains still remained one pCas9 plasmid and one gRNA plasmid. The curing system does not eliminate all plasmids, but introduces another gRNA plasmid to replace the previous gRNA plasmid. Compared to our study, we both used CRISPR technology to remove the plasmids. However, they used a gRNA plasmid to remove the original gRNA plasmid, but the final cells retained the pCas plasmid. Instead, we introduced RNA fragments to eliminate the CRISPR-Cpf1 plasmid, and we ended up with cells that were completely devoid of the plasmids.

For multi-round genome editing, such as the Cas-3P system, a complex system containing an inducible sgRNA or crRNA system and a programmed gRNA removal system has also been developed in various microorganisms. Two plasmids are generated to separately express Cas9 and sgRNA in *Bacillus subtilis*. The sgRNA plasmid contains two sgRNAs and a donor DNA for homology directed repair. The sgRNA_ct_ is chromosome-targeting under the control of a constitutive promoter, and the sgRNA_st_ is self-targeting under the control of a sporulation-specific promoter. The sgRNA plasmid can be cured in the sporulation phase after editing the chromosome in the growth phase. Therefore, the plasmid-cured cell can undergo the next-round of genome editing [30]. An all-in-one plasmid CRISPR-Cas9 system is constructed in *B. subtilis*. The plasmid contains a Cas9 expression cassette, and expresses gRNA_rep_ targeting the replication origin of the plasmid, and constitutively expresses gRNA_target_ for cleavage at the target gene. The sgRNA is self-targeting to the replication origin under the control of a rigorous acetoin-inducible promoter, which has no basal expression levels in the presence of phosphotransferase system (PTS) sugars. The system can be rapidly applied for iterative genome editing [31]. The gRNA, driven by an arabinose-inducible promoter PBAD, targets the antibiotic resistance *bla* gene and guides Cas9 to cleave the gRNA-containing plasmid for plasmid curing in *E. coli* [24]. The gRNA with the *lacI*^q^-Ptrc promoter targets the pMB1 replication origin of the plasmid for plasmid curing in *E. coli* [25]. The temperature-sensitive origin could be introduced for an efficient exclusion of the vectors. A universal plasmid-curing system (pFREE-RK2) based on the CRISPR-Cas9 approach uses gRNA driven by a rhamnose inducible promoter P_rha_, targeting ColE1/p15A, and a pSC101 replicon vector sequence to facilitate plasmid curing. The pFREE-RK2 also contains a temperature-sensitive, low-copy replicon that replicates in a broad representation of Gram-negative bacteria [18]. However, an efficient and convenient CRISPR-Cas-mediated plasmid-curing system in *S. cerevisiae* has not yet been reported. In this study, we neither engineered an inducible crRNA system nor constructed a temperature-sensitive promoter system. Only the crRNA fragments, targeting the replication origin CEN/ARS of the plasmid (Figure 1a), were transformed into cells to cleave the replication origin by the Cpf1 endonuclease, resulting in plasmid elimination from the cells (Figure 2 and Figure 3).

In addition to multiple rounds of genome-editing through an iterative process, CRISPR-Cas systems have been explored for the curing of plasmids containing antibiotic resistance genes. Antimicrobial resistance is a global problem, and antimicrobial resistance genes are frequently located on plasmids [23]. The approach of the CRISPR-Cas system was developed to remove resistance plasmids and sensitize the recipient bacteria to antibiotics. A mobilized colistin resistance (*MCR-1*) gene confers plasmid-mediated resistance to colistin, one of several last-resort antibiotics for treating Gram-negative infections. One strategy, a conjugative CRISPR-Cas9 system for the removal of *MCR-1*-carrying plasmids from *E. coli*, provides a new way to counteract the gradual spread of *MCR-1* in bacterial pathogens [32]. A CRISPR-Cas9-mediated plasmid-curing system (pCasCure) was developed and electrotransferred into various clinical carbapenem-resistant *Enterobacteriaceae* (CRE) isolates. This platform is highly efficient in the elimination of epidemic carbapenem-resistant plasmids [33]. Moreover, many microorganisms have been found to carry native cryptic plasmids. The CRISPR-Cas9 system is also applied to eliminate native plasmids. *Leuconostoc citreum* CB2567 isolated from fermented kimchi contains a cryptic plasmid that is difficult to eliminate. The CRISPR-Cas9 system is applied to remove cryptic plasmids by transforming a Cas9-expressing killer plasmid with a gRNA targeting specific cryptic plasmid DNA, and developing a safe food-grade host strain without residual antibiotic markers [34]. There are five native high-copy plasmids over 30 kb in the efficient ethanol-producing bacterium, *Zymomonas mobilis* ZM4. Two native plasmids were eliminated by the CRISPR-Cas9 system, and there was no obvious effect on the growth of the mutant strain [35]. The CRISPR-Cas9 system is introduced to eliminate native plasmids from *Bacillus anthracis* and *Bacillus cereus*. The plasmid elimination rate is higher for sgRNAs targeting the origin of replication (96–100%) than the non-replication initiation region (88–92%) [36]. The application of plasmid-curing technology based on the CRISPR-Cas system in this study is not only in the multi-rounds of genome-editing, but also in the antimicrobial resistance for plasmid curing.

In the CRISPR genome editing system, introduced DNA serves as a template for homologous recombination during the process of DNA repair [10]. In our study, the synthesized ssDNA (CEN HA) was used as a homologous recombination template, containing 50 bp DNA upstream and downstream of the replication origin CEN/ARS. During the genome editing process, the DNA sequence at the origin of replication was cleaved by the complex of crRNA/Cpf1. The ssDNA serves as a crucial DNA template, allowing for the repair of the cleaved DNA through homologous recombination, resulting in the removal of the origin of replication. When the plasmid lacks an origin of replication, it becomes incapable of replicating. Therefore, disruption of the replication origin of CRISPR plasmids results in the plasmid curing in *S. cerevisiae* (Figure 1). Furthermore, without introducing the ssDNA (CEN HA), the DNA sequence at the origin of replication was still cleaved by the complex of crRNA/Cpf1. The cleaved DNA could not be repaired through homologous recombination. However, the replication origin sequence was cleavage, and the replication cannot proceed unless the DNA is spliced back (Figure 2). The efficiency of the plasmid curing remained above 70%, introducing only crRNA fragments without an ssDNA template. It is demonstrated that once the replication origin sequence of the plasmid is destroyed, the plasmid can be eliminated.

Off-target effects in the CRISPR genome editing system can occur when the Cas protein and gRNA mistakenly recognize and cleave DNA sequences that closely resemble the intended target site. It happens and is related to the designed gRNA or crRNA sequences [37]. The sequence of the crRNA, crCEN1 used in this study, was subjected to BLAST analysis against the genomic sequences of *S. cerevisiae* S288C. The identical sequences on chromosome II were observed. However, it does not affect the growth of the strain. To reduce the risk of off-target effects, alternative crRNA sequences targeting to the replication origin CEN/ARS can be designed in the future.

Electroporation is the process of exposing cells to an electric field that causes the plasma membrane to become more permeable. The increased permeability is only temporary, and after a certain period of time, the cells repair their plasma membranes and return to homeostasis. Cell recovery is an active cellular process that reduces cell membrane damage after electroporation [38]. In a previous study, we observed longer recovery times for colony development after electroporation and higher genome editing efficiency in the CRISPR-Cpf1 system in *S. cerevisiae*. The efficiency of genome editing is highest at 48 h for cell recovery [27]. However, the curing efficiency did not consistently increase with cell recovery time in this study. The curing efficiency was increased for 5 h cell recovery after electroporation, and it was constant for 48 h cell recovery after electroporation (Figure 2c). The curing efficiency was highest within 5 h cell recovery periods, but not increased for longer recovery periods. The reason might be that RNA is highly unstable in vivo due to the presence of ribonucleases. In the absence of Cpf1 protein, crRNAs are unstable and rapidly degraded in 5 h in vivo [39].

In summary, we have developed a CRISPR-mediated plasmid-curing system. The plasmid-curing system introduced a crRNA targeting the replication origin of the plasmid and a single-strand DNA for homologous recombination. The efficiency of the plasmid curing was up to 96%. We also developed a simplified plasmid-curing system by introducing only one crRNA fragment. This plasmid curing strategy can be applied in broad hosts by designing crRNA specific for the replication origin of the plasmid. The plasmid curing system via CRISPR-Cas editing technology has multiple applications, such as multi-rounds of genome-editing through an iterative process, as a commercial product without traceable foreign genes, and even the elimination of native plasmids in microorganisms.

## Figures and Tables

**Figure 1 biomolecules-13-01561-f001:**
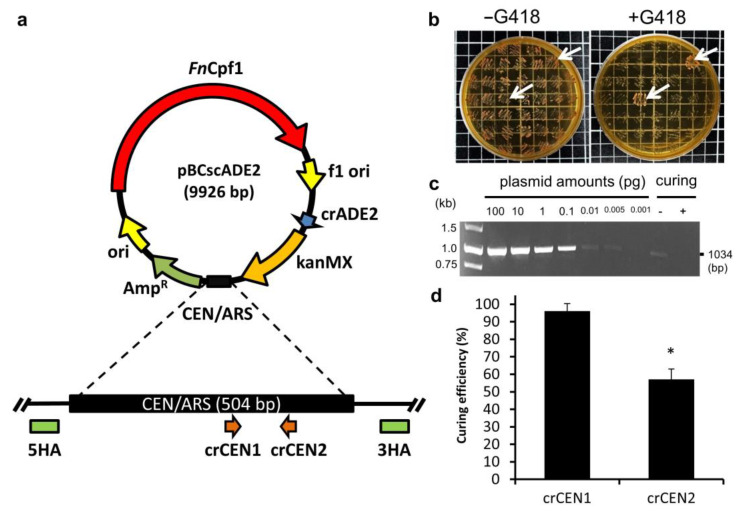
Development of a plasmid curing system using CRISPR-Cpf1 system in *S. cerevisiae*. (**a**) Schematic representation of the plasmid pBCscADE2, crRNAs targeting for the replication origin and the homologous DNA template of CEN/ARS sequence. (**b**) Plasmid-cured colonies can grow on YPD medium without G418, but not on medium containing G418. The arrows indicate the colonies which failed plasmid curing. (**c**) Colony PCR by primers specific for the plasmid. (**d**) The plasmid-curing efficiency for crCEN1 and crCEN2. Strains were transformed crCEN1 or crCEN2 with CEN HA donor DNA and plated on YPD medium with or without G418. Data were derived from three separate experiments and are presented as the means ± SD. The symbol * indicated statistically significant differences (*p* < 0.05 using Student’s *t*-test).

**Figure 2 biomolecules-13-01561-f002:**
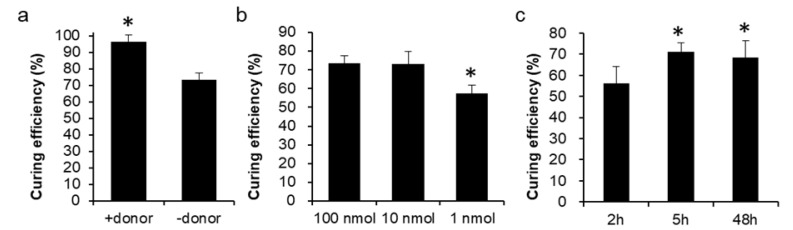
Optimal conditions for the plasmid-curing efficiency. (**a**) The plasmid-curing efficiency that transformed crCEN1 with or without CEN HA. (**b**) The plasmid-curing efficiency that transformed different concentrations of crCEN1. (**c**) The plasmid-curing efficiency that recovered for different periods after electroporation. Data were derived from three separate experiments and are presented as the means ± SD. The symbol * indicated statistically significant differences (*p* < 0.05 using Student’s *t*-test).

**Figure 3 biomolecules-13-01561-f003:**
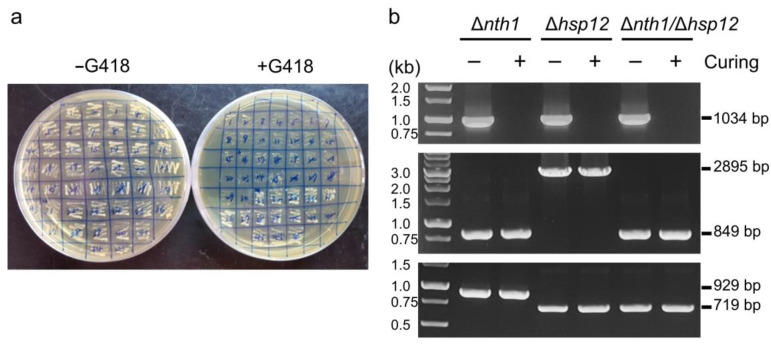
The Δ*nth1*, Δ*hsp12*, and Δ*nth1*/Δ*hsp12* strains for plasmid curing. (**a**) The colonies selected for the antibiotic susceptibility tests. (left plate) YPD medium and (right) YPD medium with G418 antibiotics. (**b**) Colony PCR analysis for Δ*nth1* strain, Δ*hsp12* strain, and Δ*nth1*/Δ*hsp12* strain with or without plasmid curing. DNA was amplified by colony PCR, the PCR product size were 1034 bp for the plasmid backbone DNA (upper), 2895 bp for *NTH1* and 849 bp for truncated *NTH1* (middle), and 929 bp for *HSP12* and 719 bp for truncated *HSP12* (bottom). Lane M, DNA ladders, land −, strains without plasmid curing, land +, plasmid cured strains.

**Figure 4 biomolecules-13-01561-f004:**
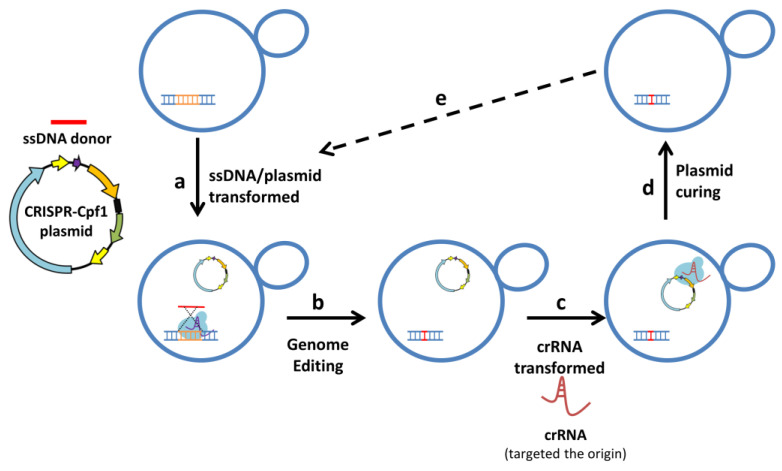
The multi-round CRISPR-Cpf1 genome-editing system in *S. cerevisiae*. (**a**) Transformation of the CRISPR-Cpf1 plasmid and single-strand DNA donor into yeast. (**b**) The target gene is edited by CRISPR-Cpf1 system. (**c**) Transformation of the crRNA targeting the replication origin of the CRISPR-Cpf1 plasmid into the edited yeast. (**d**) The replication origin of the plasmid is cleaved by Cpf1, resulting in the plasmid being cured. (**e**) The yeast is ready for the next cycle of gene editing.

**Table 1 biomolecules-13-01561-t001:** Targeted sequences, primers and homologous templates of the replication origin CEN/ARS used in this study.

crRNA	Targeted Sequence	PAM	Primer Sequence for Colony PCR	ssDNA Template Sequence
crCEN1	TTT CGT GTG TGG TCT TCT ACA CAG ACA A	TTTC	F1-F: TTT TTC GCC CTT TGA CGT TG KX-R: ATC GCG AGC CCA TTT ATA CC	CEN HA: GAG ACG AAA GGG CCT CGT GAT ACG CCT ATT TTT ATA GGT TAA TGT CAT GAT ATT TGT TTA TTT TTC TAA ATA CAT TCA AAT ATG TAT CCG CTC ATG AGA C
crCEN2	TTT CCG AAG ATG TAA AAG ACT CTA GGG G	TTTC		

## Data Availability

The data presented in this study are available on request from the corresponding author.

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
