# Peer review of "An Introduced RNA-Only Approach for Plasmid Curing via the CRISPR-Cpf1 System in Saccharomyces cerevisiae"

_biomolecules, 2023, doi:10.3390/biom13101561_

Round 1

Reviewer 1 Report

The manuscript is interesting and, in terms of the English language, it is perfect. However, some points must be improved:

1) Line 27: "Plasmids MAY BE independently replicated", instead of "ARE".

2) In the introduction, define what "cr", "sg", "tracr", and "g", along with RNA, stand for. Please do the same for Cpf1, PAM, and Cas9. 

3) Lines 146-148: please rewrite both sentences in just one. And please elucidate why NTH1- and HSP12-deleted strains were used. It seems that the ADE2-deleted strain was already enough to make the authors' point. So, explain why these strains were chosen. Please present some data about these deleted genes in the introduction so these deletions make sense to the readers.

4) Please elucidate the sequences of the crRNAs. Are they the same as those pointed as oligonucleotides in Table 1?

5) Please state that KanMX cassette is a resistance gene for G418 Geneticin.

6) Although the second strategy (crRNA-only/alone) was clearly described in the manuscript, the first strategy must be better elucidated. It is not clear what "CEN HA" stands for. What is the role of the single-strand DNA? Was it employed instead of the crRNA alone? This must be clarified in the manuscript.

Reviewer 2 Report

In the present study, the authors have reported the development of a plasmid-curing system utilizing CRISPR-Cpf1 technology in S. cerevisiae. Following the editing of strains with the CRISPR-Cpf1 plasmid, a crRNA fragment targeting the CRISPR plasmid was subsequently introduced to facilitate the removal of said plasmid, resulting in a plasmid-curing efficiency of approximately 70%. This technique holds significant implications for applications such as multi-round CRISPR-Cpf1 genome editing in S. cerevisiae and the elimination of foreign genes for commercial products. The authors provide comprehensive introductions to yeast plasmids, CRISPR-Cpf1, as well as the importance and prior endeavors related to plasmid curing. The results are clearly presented, and the multi-round CRISPR-Cpf1 genome editing system is effectively illustrated in Figure 4.

Several aspects of this work could benefit from improvement. Firstly, in the context of actual multi-round CRISPR-Cpf1 genome editing experiments, is it imperative to eliminate the CRISPR-Cpf1 plasmid? Could we simply introduce crRNA and template ssDNA for subsequent rounds of gene editing, given that this study has already demonstrated its efficacy in gene editing? Secondly, while the authors have thoughtfully reviewed prior efforts related to plasmid curing within the CRISPR-Cas system, these efforts are also reiterated in the discussion section, resulting in some redundancy between the introduction and discussion segments. Moreover, could the authors please provide comments on the potential off-target effects of this system? Lastly, the authors should provide more details for the methods section, given that this is a technique-oriented research, and especially because some of the references (like ref 38) are not freely accessible.

The quality of English could be enhanced if the authors incorporate more transitional words and phrases between sentences to better convey their reasoning and improve overall clarity.

Reviewer 3 Report

The authors have demonstrated a CRISPR/CpfI based plasmid curing system in S. cerevisiae. The topic is really interesting to the readers since plasmid curing is an important step in strain development for commercial purposes and having an efficient curing method can save lot of time and effort for passaging strains multiple times through non-selective conditions. However, the manuscript has not covered the detailed explanation on sequences used and the methodology adopted to perform experiments. These concerns are covered in detail below:

a) I completely failed to understand concept of the transformation of crRNA in this manuscript which is also highlighted in figure 4. Does the author claims to directly transformed the RNA instead of DNA?

b) If authors transformed ssDNA then there is lot of uncertainty regarding the sequences used for crRNA design and the regulatory elements. For example, it is not sure which promoter was used to initiate transcription to produce crRNA. How the fragment survived on its own without integrating.

c) The authors need to provide more detail on their emphasis on Introduced RNA only approach for plasmid curing.

d) Is this the first time RNA fragments were directly used in S. cerevisiae (or in Bacteria) instead of gRNA plasmid? 

e) The material and methods need to be more detailed with emphasization on sequences and each step for replication of results by other authors to adopt this system.

f) Introduction and discussion are similar at most parts covering the same information and same literature.

g) The provided data is very limited to support all the claims. If this is the first time use of CRISPR/Cas9 based plasmid curing in S. cerevisiae. It needs to be highlighted as a proof of concept study.

Round 2

Reviewer 3 Report

The authors have provided adequate responses to the questions raised by me in my previous review. I am satisfied with the current version. Therefore, I endorse the manuscript for publication in its current form.